

# Registered report: investigating a preference for certainty in conversation among autistic adults compared to dyslexic adults and the general population

Alexander C. Wilson and Dorothy V.M. Bishop

Department of Experimental Psychology, University of Oxford, Oxford, UK

## ABSTRACT

Social communication difficulties are a diagnostic feature in autism. These difficulties are sometimes attributed, at least in part, to impaired ability in making inferences about what other people mean. In this registered report, we test a competing hypothesis that the communication profile of adults on the autism spectrum can be more strongly characterised by reduced confidence in making inferences in the face of uncertain information. We will test this hypothesis by comparing the performance of 100 autistic and 100 non-autistic adults on a test of implied meaning, using a test of grammaticality judgements as a control task. We hypothesise that autistic adults will report substantially lower confidence, allowing for differences in accuracy, than non-autistic adults on the test of implied meaning compared to the grammaticality test. In addition, we hypothesise that reduced confidence in drawing inferences will relate to the cognitive trait Intolerance of Uncertainty and self-reported social communication challenges. Finally, we will conduct exploratory analysis to assess the specificity of the communication profile of the autistic adults by comparing their performance to that of dyslexic adults, who might also be expected to experience challenges with language and communication.

## INTRODUCTION

Persistent challenges with social communication are a defining feature for the diagnosis of autism (*American Psychiatric Association, 2013*). The underlying nature of these challenges remains unclear, although they are sometimes attributed to a core impairment in pragmatics (*Baron-Cohen, 1988*; *Rapin & Dunn, 2003*). Pragmatics refers to the role of context in communication, including the ability to 'read between the lines' to infer intended meaning beyond what is explicitly stated (*Baird & Norbury, 2016*). However, empirical research suggests that pragmatic difficulties may be rather subtle in autistic people, and mostly attributable to language ability (*Kalandadze et al., 2018*; *Loukusa & Moilanen, 2009*). An alternative suggestion is that social communication difficulties are less the result of an impairment in pragmatics, but more impacted by cognitive preferences that differ between autistic and non-autistic people. We propose that a *preference for*

Corresponding author
Alexander C. Wilson,
alexander.wilson2@psy.ox.ac.uk

*certainty and explicit communication* commonly occurs in autistic people, and that this trait may be a critical factor in the communication difficulties experienced by autistic people, as communicative situations often involve ambiguity, uncertainty and implied meanings.

## Intolerance of uncertainty

A *preference for certainty and explicit communication* may link to the widely-researched cognitive trait, Intolerance of Uncertainty (IU). IU has been defined as a tendency to negatively evaluate uncertain situations and information (*Shihata et al., 2016*). We use the term 'Intolerance of Uncertainty' in line with previous research and intend to convey a value-neutral meaning in using it, as we recognise that high levels of IU may be an understandable, even adaptive, response where individuals have experienced mishaps in confusing situations. IU has mostly been investigated as a transdiagnostic construct that plays a central role in emotional disorders across the general population (see *Shihata et al. (2016)* for a review), but it also seems especially relevant to autism, with autistic children and adults showing significantly elevated levels of the trait compared to the general population (*Hwang et al., 2020*; *Vasa et al., 2018*). IU has been closely linked to anxiety in autistic people (*Jenkinson, Milne & Thompson, 2020*), and also relates to core features of autism, including social difficulties, sensory sensitivities, insistence on sameness and repetitive behaviours (*Hwang et al., 2020*; *Vasa et al., 2018*; *Wigham et al., 2015*).

A possible link between IU and communication in autistic people remains largely unexplored. However, there are reasons to believe that a link is plausible. First, inferential models of communication, such as Relevance Theory (*Sperber & Wilson, 1986*), propose that communication inherently involves uncertainty. Under Relevance Theory, language comprehension is not simply a process of 'understanding what the words mean', as there are often indeterminacies and ambiguities in uses of language. Instead, the words are used as evidence by the listener in supporting a hypothesis about what the speaker *probably* means in the context—that is, the listener infers the speaker's intended meaning under conditions of uncertainty. Relevance Theory suggests that there is a gradient of uncertainty in communication. Sometimes, the listener can rely mostly on the explicit content of the utterance to compute the intended meaning. In other situations, there is a greater reliance on inferential processing to understand the speaker's probable meaning by integrating the utterance with contextual cues and world knowledge. Compare for instance the utterances 'No, let's stay inside' and 'It's quite cold today' as responses to a suggestion to go outside. In the second example, the speaker communicates implicitly, leaving the listener to process the implicature that they would probably prefer to stay inside. In a previous study, we provided evidence for cognitive differences between autistic and non-autistic people in processing implicatures (*Wilson & Bishop, 2020b*). Crucially, it seemed that a cognitive preference for certainty and explicit communication, and not simply reduced ability, may account for some of the differences. Participants completed the Implicature Comprehension Test, which required individuals to listen to short
conversational interchanges that are followed by a comprehension question to assess whether an implied meaning has been processed; test-takers responded with 'yes', 'no' or 'don't know'. Controlling for grammar/vocabulary ability, we found that autistic adults ($N = 66$) were 6.19, 95% CI [3.63–10.39], times more likely to select the 'don't know' rather than the correct response compared to non-autistic people ($N = 118$), and also 2.56, 95% CI [1.76–3.77], times more likely to choose the 'incorrect' response (*Wilson & Bishop, 2020b*). Group differences were large, and performance on the test gave 76% sensitivity and specificity for differentiating between autistic and non-autistic groups. On the face of it, these results suggest that autistic people have difficulties inferring the gist of a speaker's meaning, as predicted by the 'central coherence' theory. This theory proposes that autistic people may show less tendency than non-autistic people to process information at a global level (*Frith, 1989*).

However, in an alternative version of the test without a 'don't know' response, autistic individuals showed high accuracy for items for which they had selected 'don't know' first time round. This marked tendency to select 'don't know' when given a chance, but to process the inference as intended when constrained by the task, suggested reduced confidence in the face of uncertain information and a preference for explicit communication. This could be due to possible difficulties around metacognition in autistic people, who may experience a mismatch between performance and confidence in their performance due to differences in self-monitoring. There is evidence that autistic people may show such a mismatch (*Grainger, Williams & Lind, 2016*; *Nicholson et al., 2019*) although there is some concern about the replicability of these results (*Maras, Norris & Brewer, 2020*). An alternative view would be that it is less an issue of metacognitive 'ability', and more about differences in personality/cognitive preference, with the well-replicated elevated levels of IU in autistic people (*Hwang et al., 2020*; *Vasa et al., 2018*) accounting for this apparent preference for explicit communication observed in our previous study. In the present study, we aim to replicate this finding with more refined methods. In an adapted version of the Implicature Comprehension Test, individuals will respond using a 4-point scale of 'yes', 'maybe yes', 'maybe no', and 'no', allowing us to capture accuracy and confidence in the same measure. We hypothesise that confidence is likely to be affected specifically in a pragmatic language task (i.e. where the individual needs to make flexible context-dependent inference about uncertain implied meanings), and not on tasks focused on more structural, codified aspects of language such as grammatical competence. As such, we present the Grammaticality Decision Test as a control task with a similar response format to the pragmatic task to test the specificity of any differences.

We propose that reduced confidence on the Implicature Comprehension Test will be a marker of IU in autistic people, and may be a more influential factor in the communication difficulties diagnostic of autism, as opposed to a 'deficit' in understanding social meanings. If this claim is borne out, it would have a couple of implications for psychological practice. First, it would suggest that interventions targeting IU may be useful for autistic people wanting support with communication challenges. Current interventions

for communication focus on explicit instruction in social skills, and reviews suggest modest effectiveness although there are questions about the extent to which skills transfer to daily life (*Gates, Kang & Lerner, 2017*; *Spain & Blainey, 2015*). A focus on IU may be a useful alternative target. Existing cognitive interventions involve integrating psychoeducation and cognitive challenge techniques to target a client's beliefs about (un)certainty, and these have shown some effectiveness for treating mental health difficulties and particularly anxiety in the general population (*Shihata et al., 2016*). It remains to be seen whether such interventions could be adapted to support autistic individuals with distressing communication experiences, although this is a promising possibility given that early studies suggest that such interventions may be feasible and acceptable for autistic groups (*Rodgers et al., 2018*, *2017*). Second, if a cognitive preference for certainty is especially significant as an explanation for social difficulties, then it supports an autism-positive approach to intervention which focuses on awareness of cognitive differences across communities. In addition, if performance on the Implicature Comprehension Test is a sensitive marker of IU, that in itself might have clinical and research utility, since measurement of IU is currently limited to self- and informant-report questionnaires.

A remaining question is whether any differences observed on our tasks are specific to autism or might also be relevant to other neurodevelopmental diagnoses. This is certainly plausible in the light of dimensional models of neurodiversity, where features of autism, developmental language disorder, dyslexia, ADHD, etc., might show some overlap and exist as a continuum in the general population (*Thapar, Cooper & Rutter, 2017*). To test the specificity of any cognitive differences observed on our tests, we will compare performance by autistic people to both a dyslexic and a general population sample. As neurodevelopmental conditions are often co-occurring, we view these three groups as defined less by a specific diagnostic label but rather as varying along a communication continuum. As such, one group is defined by social communication differences potentially alongside co-occurring language/literacy impairments (the autism group), a second group by language/literacy impairments but no diagnosed social communication difficulties (the dyslexia group), and a final group without any communication, language or literacy related diagnosis. It is possible that dyslexic adults may show some difficulty on our pragmatic task (i.e. the Implicature Comprehension Test), as previous research has documented some limited evidence for pragmatic difficulties in dyslexic individuals (*Cappelli et al., 2018*; *Cardillo et al., 2018*; *Griffiths, 2007*). An alternative possibility is that adults with dyslexia will show greater difficulty with tasks focused more on structural language skills compared to pragmatics. For instance, a meta-analysis has found that dyslexic adults perform less well on language measures, such as vocabulary, speeded naming, verbal memory and phonological processing, than people without a diagnosis of dyslexia, with moderate to large effect sizes (*Swanson & Hsieh, 2009*). Given that there is no clear reason to support one of these possibilities over the other, we will take a more exploratory approach with the dyslexic group to examine how they compare with autistic adults.

In summary, we propose the following hypotheses:

1. Autistic adults will score lower on a pragmatic language task when responses are coded purely in terms of confidence (number of yes and no responses, regardless of polarity) than when responses are coded in terms of accuracy (with yes and maybe yes, and maybe no and no responses, combined according to polarity), compared to adults without any neurodevelopmental diagnosis, but will not show this same disparity between accuracy and confidence on a core language task.
2. The number of less confident responses (maybe responses) on the pragmatic language task, the score on the Intolerance of Uncertainty Scale, and self-reported social communication difficulties will significantly intercorrelate across the full sample.

## METHODS

Ethical approval for this project was granted on 30/03/2020 by the Medical Science Interdivisional Research Ethics Committee at Oxford University (Ref: R68518/RE001). The script for the power analysis and example materials are available on the Open Science Framework here (https://osf.io/wk97s/). We are very happy to share full materials for this study, but to protect the validity of the items for future uses, we ask that researchers contact us to request a link to the full assessments. See further information on requesting access by following the link above.

### Power calculation

We determined power to detect the three-way interaction described in Hypothesis 1 using simulations. We used data reported in *Wilson & Bishop (2020b)* to estimate the likely size of fixed and random effects in the mixed model described in "Data analysis" below. Using R package simr (*Green & MacLeod, 2016*) we ran 1,000 simulations with a sample size of 200 people (100 autistic, 100 non-autistic) and a significant three-way interaction was found in 9,830 simulations, indicating that power was over 98% to detect our effect of interest at an alpha level of 0.05. Effectively, this allows us to detect a significant difference where approximate Cohen's *d* values in favour of the non-autistic group are 0.70 and 1.10 for the implicature accuracy and confidence variables and 0.20 for the grammar variables, as suggested by our previous data. Allowing for exclusion of up to 10% of participants during the outlier exclusion phase described in "Data analysis", power remains very high (98% in a sample of 180). For hypothesis 2, a sample of 200 is powered at over 99% to detect a correlation of 0.3.

### Participants

We will recruit individuals with autism, dyslexia, and no neurodevelopmental diagnosis. Based on the power calculation, we will recruit 100 autistic adults and 100 adults without a neurodevelopmental diagnosis in order to run the confirmatory analysis. In addition, we will aim to recruit 50 dyslexic adults as a clinical control group for exploratory analysis. All participants will meet the following eligibility criteria: (i) age of 18 years or over, (ii) native-level fluency in English, (iii) no history of acquired brain injury,

(iv) no significant uncorrected sensory impairment, and (v) access to a computer with internet and audio.

Individuals will be recruited into three groups defined by communication and language/literacy problems. One group will be recruited on the basis of a clinical diagnosis of autism; participants will need to declare where, by whom and what label was used for their diagnosis on the Study Questionnaire. For inclusion, the diagnosis must have been made in a clinical service by appropriately trained individuals, such as clinical psychologists, psychiatrists or developmental paediatricians. We will recruit autistic individuals through Autistica, the research network for families and individuals with autism, as well as support groups arranged privately and by the National Autistic Society, and through social media. A second group will include individuals reporting dyslexia or specific reading difficulties. For inclusion in this group, individuals must score below the clinical threshold of six on the Autism Spectrum Quotient (AQ) and at six or above on the Reading Scale of the Adult Reading Questionnaire (ARQ); this latter score translates to over 1.5 SDs above the mean in individuals not self-reporting dyslexia in the original validation study (*Snowling et al., 2012*). Individuals will be recruited through charitable organisations such as the British Dyslexia Association and social media. Other neurodevelopmental diagnoses will be noted but will not be grounds for exclusion from these groups. A third group will have no neurodevelopmental diagnosis, and will be recruited via the online participant platform, Prolific (https://prolific.co). Individuals will be excluded from this third group if they score above threshold on either the AQ or ARQ (i.e. above six on either) and if they have ever been diagnosed with: a global or specific learning disability, attention deficit hyperactivity disorder, dyspraxia/developmental coordination disorder, a genetic variation (such as Down's syndrome or Fragile-X) or a neurological condition (such as epilepsy).

## Procedure

The study will be presented online using Gorilla, the online platform for behavioural experiments and surveys (https://gorilla.sc/). Individuals complete an online set of tasks and questionnaires in one sitting at a time and place of their choosing. After providing informed written consent to participate, individuals will complete a Study Questionnaire (please see the OSF link above) on which they will be asked to report on demographics and any neurodevelopmental diagnoses. Then they will complete questionnaires/tasks in two sections. The first section will include the experimental tasks required for the hypothesis-testing, and the second will include several brief measures for characterising the sample. The two experimental tasks will be randomised between participants, and all other measures will be administered in the order set out below.

## Measures

### Section 1: experimental tasks assessing ability and confidence with pragmatics and core language

**Implicature Comprehension Test-2 (ICT-2).** In this test of pragmatic language comprehension, participants complete an adapted version of the Implicature

Comprehension Test (*Wilson & Bishop, 2019*). There is a sequence of 56 videos, each approximately 8 s long, consisting of a conversational adjacency pair between two characters: the first character asks a closed question (eliciting a 'yes' or 'no' answer) and the second character produces a short answer but does not say yes or no. Each utterance is between 5 and 10 words in length, grammatically simple, and age of acquisition of the words does not exceed middle primary school level. Following the dialogue, the participant hears a comprehension question directly based on the structure of the first character's question. The participant answers the question on a 4-point scale (yes, maybe yes, maybe no, no) by clicking buttons arranged horizontally on the screen. This is a timed task, with a time limit of 10 s for a response from the offset of the question. There are two item types: implicature and explicit-response. Utterance length and psycholinguistic variables (word frequency, word age-of-acquisition and word concreteness) are controlled across the two item types.

For 40 videos, the second character's answer is indirect, and the participant needs to process implicature to answer the comprehension question appropriately. Example:

Character 1: Did you hear what the police said?

Character 2: There were lots of trains going past.

Comprehension Question: Did he hear what the police said?

Answer: No

Half of the comprehension questions are correctly answered by 'yes' and half by 'no'. There are two measured variables: total accuracy (collapsing yes and maybe yes, and maybe no and no responses, according to polarity) out of 40 and total confidence (number of yes and no responses, regardless of polarity) out of 40.

Alongside the implicature items, there are 16 explicit-response items where the second character's answer is more explicit. In these items, the speaker intends to convey uncertainty explicitly, whereas in the implicature items, the uncertainty is in the mind of the listener. Example:

Character 1: Will we get there by seven?

Character 2: Mmm, yes maybe, I think we're near.

Comprehension Question: Will they get there by seven?

Answer: Maybe yes

For these items, the comprehension questions will encourage the participant to use the full scale, with four questions each correctly answered by 'yes', 'maybe yes', 'maybe no' and 'no'. There is one measured variable: total accuracy out of 16.

**Grammaticality Decision Test (GDT; based on *Wilson & Bishop (2019)*).** In this test of core language ability, participants listen to a sequence of 50 sentences and decide if the sentence is grammatical and well-formed or not. Half the sentences are grammatical. Grammatical violations represent mistakes that native speakers would not tend to make, such as using an incorrect verb form (e.g. I went out after I have eaten dinner) or

atypical placing of adverbs (e.g. If you can't find it, I can send again the letter). Participants are asked whether the sentences are grammatical, indicating 'yes', 'maybe yes', 'maybe no' and 'no' as their answer by clicking buttons arranged horizontally on the screen, as in the ICT-2. After offset of the sentence, participants have 10 s to give their response.

### Section 2: questionnaires and tasks for characterising the sample

**Autism Spectrum Quotient-10 (AQ-10;** *Allison, Auyeung & Baron-Cohen, 2012*). Autistic traits will be measured using this 10-item version of the Autism Spectrum Quotient (AQ). In the original validation study, the measure had 85% correct discrimination between almost 450 autistic adults and over 800 control adults. *National Institute for Health and Care Excellence (2012)* recommend use of the questionnaire for identifying individuals for comprehensive autism assessment. A clinical cut-off of six or more is taken as indicating possible autism.

**Communication Checklist Self-Report (CC-SR;** *Bishop, Whitehouse & Sharp, 2009*). This is a norm-referenced questionnaire measuring self-reported communication challenges. Participants will be presented with the pragmatic language scale (22 items). For each item, participants identify how frequently certain communication behaviours apply to them on a 4-point scale from 'less than once a week (or never)' to 'several times a day (or all the time)'. An example item is 'People tell me that I ask the same question over and over'.

**Adult Reading Questionnaire (ARQ) Reading Scale (***Snowling et al., 2012***).** Self-reported reading difficulties will be measured using this 5-item questionnaire. In the original validation study, it showed good construct validity (correlating with observed literacy ability at −0.67) and, along with self-reported dyslexia status, discriminated with 88% accuracy in identifying those with weaker literacy skills.

**International Cognitive Ability Resource (ICAR) Sample Test (***Condon & Revelle, 2014***).** This is an open-access test of general cognitive ability, which requires participants to complete 16 items across four item types: matrix reasoning, verbal reasoning, three-dimensional rotation, and letter-number sequences. The ICAR sample test has good internal consistency (alpha = 0.81), and good convergent validity (correlating at approximately 0.8 with commercial IQ measures when correcting for reliability and restriction of range; *Condon & Revelle, 2014*; *Young & Keith, 2020*).

**Synonyms Test (***Wilson & Bishop, 2019***).** General verbal ability will be measured using this 25-item test of vocabulary knowledge. Participants select which of five written words is synonymous with a target word, under a 12 s time limit. The original version of the GDT and this task showed a moderate correlation in both autistic and non-autistic samples, suggesting they are overlapping measures of core language ability (*Wilson & Bishop, 2019*, *2020a*).

**Intolerance of Uncertainty Scale (IUS-12;** *Carleton, Norton & Asmundson, 2007*). In this self-report measure of intolerance of uncertainty, participants are presented with 12 statements about uncertainty, ambiguous situations, and the future. They rate how closely each statement relates to them on a 5-point scale from 'not at all characteristic of

**Table 1  Planned analyses.**

| Research question | Hypothesis | Statistical analysis | Power analysis |
|---|---|---|---|
| Do autistic people show reduced confidence in understanding implied meanings in conversation? | Autistic adults will score lower on the Implicature Comprehension Test-2 when responses are coded in terms of confidence (number of yes and no responses, regardless of polarity) than when responses are coded in terms of accuracy (with yes and maybe yes, and maybe no and no responses, combined according to polarity), compared to adults without any neurodevelopmental diagnosis, but will not show this same disparity between accuracy and confidence on the Grammaticality Decision Test. | A mixed model will be run including the following effects: task (Implicature Comprehension Test-2 or Grammaticality Decision Test), group (autistic or no neurodevelopmental diagnosis), and response (confidence or accuracy) as fixed effects; the interactions between these fixed effects; and participant as a random effect. The significance level of the three-way interaction will offer a test of the hypothesis. | A sample of 200 people is powered at over 98% to detect the three-way interaction. |
| Do individual differences in confidence in interpreting meaning, intolerance of uncertainty and self-rated communication difficulties inter-correlate? | The number of less confident responses (maybe responses) on the Implicature Comprehension Test-2, the score on the Intolerance of Uncertainty Scale, and self-reported social communication difficulties on the CC-SR will significantly intercorrelate across the full sample. | Pearson's correlations will be computed to quantify the relationships between these three variables across the whole sample. | A sample of 200 people is powered at over 99% to detect correlations of 0.3. |

me' to 'entirely characteristic of me'. An example item is: 'When I am uncertain, I can't function very well.'

## Data analysis

Individuals will be excluded from the dataset if they have an outlying score for either (a) accuracy on the GDT or the positive control items of the ICT-2 (b) total number of timeouts across the ICT-2 and GDT. Outliers will be defined according to the method of *Hoaglin & Iglewicz (1987)*: more than 2.2 times the interquartile range below the first quartile. In previous work, these criteria led to exclusion of approximately 5% of participants, and captured individuals scoring below approximately 50% on the GDT and 70% on the positive control items of the original version of the ICT (*Wilson & Bishop, 2020a*).

Data will be analysed in R (*R Core Team, 2019*). After exclusions, total scores on the two experimental tasks for the groups with autism and no neurodevelopmental diagnosis will be turned into long format, and each participant's total will be coded for task (ICT-2 or GDT), group (autistic or no neurodevelopmental diagnosis), response (accuracy or confidence) and participant. We will run a mixed effects linear regression using the lme4 R package (*Bates et al., 2015*). The model will include three fixed effects (task, group and response) and the interactions between these, as well as a random effect (participant). The significance level of the three-way interaction will offer a test of Hypothesis 1. We will also compute correlations between confidence on the ICT-2, self-reported

communication challenges on the CC-SR and total score on IUS-12 across the full sample; this will test Hypothesis 2 and represents a dimensional analysis of the relationship between communication difficulties and sensitivity to uncertainty. Table 1 shows a summary of our planned analyses, linking research questions, hypotheses, tests and power calculations.

In exploratory analysis, we will examine how the dyslexic group compares to the autistic group and the group without a neurodevelopmental diagnosis on the ICT-2 and GDT in terms of accuracy and confidence.

### Funding
This work is funded by the European Research Council (Ref 694189). The funders had no role in study design, data collection and analysis, decision to publish, or preparation of the manuscript.

### Grant Disclosures
The following grant information was disclosed by the authors:
European Research Council: 694189.

### Competing Interests
Dorothy V.M. Bishop is an Academic Editor for PeerJ.

### Author Contributions
- Alexander C. Wilson conceived and designed the experiments, authored or reviewed drafts of the paper, and approved the final draft.
- Dorothy V.M. Bishop conceived and designed the experiments, authored or reviewed drafts of the paper, and approved the final draft.

### Ethics
The following information was supplied relating to ethical approvals (i.e. approving body and any reference numbers):

Ethical approval for this project was granted on 30/03/2020 by the Medical Science Interdivisional Research Ethics Committee at Oxford University (Ref: R68518/RE001).

### Data Availability
As this is a registered report, there is not currently any data to submit. However, please see the OSF project linked to this RR (https://osf.io/wk97s/) where we provide the power analysis script, study questionnaire and some example items for the tasks.

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
