# Peer review of "Registered report: investigating a preference for certainty in conversation among autistic adults compared to dyslexic adults and the general population"

_PeerJ, doi:10.7717/peerj.10398_

## Round 0.1 · original submission · Minor Revisions

I have now received one independent review on your report, which you submitted to PeerJ, I had the possibility to see three reviews of a previous submission of this paper and have carefully read it myself. As you will see, the reviewer is positively oriented toward your report, and I share his/her opinion. The reviewer simply asks to make the materials available, and I also think that you could refer more extensively to literature on metacognition, directly relevant to the issue of confidence and tolerance for uncertainty. I was also wondering – but this is a personal curiosity, feel free to avoid referring to it in the revised version - whether this difficulty with uncertainty might be related to the difficulty of autistic people with abstractness. Finally, I think that the impactful question of how to intervene to improve the level of acceptance of uncertainty should be addressed.

·

Basic reporting

The article is very well written and easy to follow. Appropriate literature is cited. Hypotheses are clearly stated and follow logically from the introduction. Link to OSF given. My only major suggestion is to share the full set of items so that it is easier to understand what is being tested. It might also help to expand on what is proposed to be involved in processing an inference/ implicature by giving some theoretical background. Reference to relevant theoretical frameworks would be helpful as there is some debate about when inferences are and are not necessary and what they involve in different cases (e.g., Kintsch, 1998, Comprehension). Lack of a theoretical statement on this and lack of access to materials made me wonder what was in fact being tested.

Other notes:
If a value-neutral term is intended would 'preference for certainty' be better or is 'intolerance of uncertainty' preferred as a conventional expression?

The section in the intro on the inclusion of a dyslexic group - and the selection of this group rather than others - was a little vague at first but became clearer in the method around line 172. Perhaps this justification could be given earlier.

Experimental design

The proposed research is original primary research within the aims and scope of the journal. With the caveat above, the research question is well defined, the means of testing it are justified and the results are likely to advance knowledge. Methodological and analytic standards are high. Detail is sufficient to replicate with the exception of provision of key inference test materials. Unless there is a good reason not too, these materials should presumably be shared and this would be my only major request for revision. This is all the more important given how difficult it is to define inference, to come up with good tests of it and to be clear why they are a test of inference (perhaps even this is a question of individual differences to some extent, with some arriving at a given 'inference' more through spreading activation and others by making a more proactive effort after meaning). Of course this is work in progress and some of this has to be left underspecified for now. The purpose of sharing materials would not be to scrutinise every item before the test is run, but to help everyone to make sense of what is being tested and what is found.

Having said this, I had specific notes about some test items. For example, in the video about free seats, there are free seats available in the background of the picture so the answer is also available visually.

I also wondered whether there were specific reasons why people were likely to struggle with different items - for example, reasons relating to real world knowledge and ability to generate world models that would make the speech observed plausible. It appears this could relate to uncertainty in several ways. For example, someone could be uncertain because they struggled to generate plausible real world contexts that would justify the answer or because they could generate several scenarios and weren't sure which was more likely to hold (e.g., 'I heard the police say there were a lot of trains' or 'there were so many trains, I couldn't hear what the police said').

Other notes:
Missing word on line 179 > the online platform

Perhaps clarify that for the literal items, the speaker intends to convey uncertainty. For the implicature items, the uncertainty is in the mind of the listener.

Validity of the findings

N/A

Additional comments

I look forward to hearing about the results.

---

## Round 0.2 · accepted · Accept

I am happy to inform you that your registered report has been accepted for publication.

I am looking forward to seeing the results!

Thank you for sending your interesting work to PeerJ.